# Machine-Learning Based Detection of Coronary Artery Calcification Using Synthetic Chest X-Rays

**Dylan Saeed**[1]                  D.SAEED@STUDENT.UNSW.EDU.AU
**Ramtin Gharleghi**[3]                  R.GHARLEGHI@UNSW.EDU.AU
**Susann Beier**[2]                   S.BEIER@UNSW.EDU.AU
**Sonit Singh**[1]                   SONIT.SINGH@UNSW.EDU.AU

[1] *School of Computer Science and Engineering, University of New South Wales, Australia*

[2] *School of Mechanical and Manufacturing Engineering, University of New South Wales, Australia*

[3] *Independent Researcher, Sydney, Australia*

**Editors:** Accepted for publication at MIDL 2026

## Abstract

Coronary artery calcification (CAC) is a strong predictor of cardiovascular events, with computed tomography (CT)-based Agatston scoring widely regarded as the clinical gold standard. However, CT is costly and impractical for large-scale screening, while chest X-rays (CXRs) are inexpensive but lack reliable ground truth labels, constraining deep learning development. Digitally reconstructed radiographs (DRRs) offer a scalable alternative by projecting CT volumes into CXR-like images while inheriting precise labels. In this work, we provide the first systematic evaluation of DRRs as a surrogate training domain for CAC detection. Using 667 CT scans from the COCA dataset, we generate synthetic DRRs (posterior–anterior and lateral views per scan) and assess model capacity, super-resolution (SR) fidelity enhancement, preprocessing, and training strategies. Lightweight convolutional neural networks (CNNs) trained from scratch outperform large pretrained networks (DenseNet121, ResNet18); pairing super-resolution with contrast enhancement yields significant gains; and curriculum learning stabilises training under weak supervision. Our best configuration achieves a mean area under the receiver operating characteristic curve (AUC) of 0.754, comparable to or exceeding prior CXR-based studies. These results establish DRRs as a scalable, label-rich foundation for CAC detection, while laying the foundation for future transfer learning and domain adaptation to real CXRs.

**Keywords:** Coronary artery calcification, digitally reconstructed radiographs, deep learning, super-resolution, domain adaptation, medical imaging, chest X-ray

## 1. Introduction

Coronary artery calcification (CAC) is a well-established marker of atherosclerotic burden and a strong predictor of cardiovascular events and all-cause mortality (Budoff et al., 2007). Quantification using ECG-gated cardiac CT via the Agatston method remains the clinical gold standard for risk stratification (Agatston et al., 1990; Grundy et al., 2019). However, CT screening is costly, resource-intensive, and impractical for large-scale population use (Søgaard et al., 2022), motivating alternative modalities that are low-cost, scalable, and suitable for population-level risk assessment.

In contrast, chest X-rays (CXRs) are inexpensive and widely accessible, making them a low-cost, scalable modality for opportunistic CAC risk assessment (D'Ancona et al., 2023). Yet, CXRs have limited sensitivity for calcium depiction and lack standardised annotation protocols, leaving reliable ground truth labels scarce (Kamel et al., 2021). Importantly, while a CT scan is not required at inference time for a CXR-based model, supervised training demands ground-truth CAC labels, which can only be derived from a corresponding CT. This necessitates retrospective assembly of paired CXR–CT datasets where both modalities were acquired within a clinically acceptable window (often ≤6 months). Such dual-modality acquisitions are uncommon in routine care and, when available, can introduce temporal misalignment and label noise due to disease progression between scans. This bottleneck constrains the development of deep learning methods, and prior studies (Kamel et al., 2021; D'Ancona et al., 2023; Jeong et al., 2024) remain limited in scale and generalisability.

In such data-scarce settings, an emerging strategy is to train models within surrogate domains: synthetic or simulated approximations of the target modality that preserve key imaging physics while offering abundant labels (Unberath et al., 2018; Moore et al., 2011). In clinical domains, digitally reconstructed radiographs (DRRs) are synthetic 2D projections of CT volumes that approximate real CXRs while being able to inherit precise CT-derived labels. Prior work in their generation has validated their clinical fidelity (Moore et al., 2011), suggesting that DRRs may serve as a scalable surrogate training domain where methodological feasibility can be established before transferring to real CXRs. Although direct transfer to real CXRs is not evaluated here, recent work has shown DRR-trained models to generalize effectively in fluoroscopic and interventional settings, supporting the plausibility of similar transferability for CAC detection (Unberath et al., 2018).

To our knowledge, this work provides the first systematic evaluation of DRRs as a surrogate training domain for CAC detection. We evaluate feasibility across model capacity, fidelity enhancement, preprocessing, and training strategies, and show that DRRs constitute a scalable and precisely labelled domain for developing models prior to transfer to real CXRs, laying the foundation for low-cost, population-scale cardiovascular risk screening using existing radiography infrastructure.

## 2. Related Work

### 2.1. CT-based CAC detection

ECG-gated cardiac CT is the clinical gold standard for CAC quantification via the Agatston method (Agatston et al., 1990), and numerous studies have leveraged CT directly for automated CAC detection and scoring using deep learning (Eng et al., 2021). These approaches benefit from precise attenuation-based ground truth but remain limited by the cost, radiation dose, and infeasibility of CT screening at the population level (Søgaard et al., 2022).

### 2.2. CXR-based CAC detection

Given the wide availability of chest radiographs, several groups have explored their utility for CAC risk assessment. (Kamel et al., 2021) trained an attention-augmented VGG16 on 1,689 CXRs paired with CTs, achieving an AUC of 0.73 for CAC classification. (D'Ancona

et al., 2023) demonstrated that deep learning on CXRs could refine pretest probability estimation in suspected angina patients, validated against invasive coronary angiography. (Jeong et al., 2024) proposed a radiomics-based approach requiring manual cardiac segmentation, reporting an AUC of 0.808 for detecting moderate-to-severe CAC (>100 Agatston units). More recently, (Jeong et al., 2025) combined CXR features with multimodal patient data in a multi-objective learning framework for opportunistic CAC screening. Alternative accessible modalities have also been explored: (Song et al., 2021) and (Hsieh and Budoff, 2022) demonstrated that dual-energy chest radiography can detect and quantify coronary calcium without CT, exploiting differential X-ray attenuation across energy levels. Collectively, these approaches demonstrate the potential of low-cost radiographic modalities for CAC screening, yet progress remains constrained by the difficulty of assembling large paired datasets, weak calcification sensitivity, and dependence on manual segmentation or specialised hardware—factors that hinder scalability and reproducibility.

### 2.3. Synthetic imaging with DRRs

DRRs have been widely used in radiotherapy and orthopaedics to simulate radiographs from CT volumes, and prior work has validated their anatomical and clinical fidelity (Moore et al., 2011). Recent studies have begun to explore DRRs as surrogate datasets for training deep learning models in scenarios where paired real-world imaging and labels are scarce (Unberath et al., 2018). However, these efforts have primarily focused on registration or dose optimisation rather than disease detection. To the best of our knowledge, no systematic evaluation has been conducted on their feasibility as a surrogate training domain for CAC detection.

## 3. Methods

### 3.1. Dataset and Labels

We use the publicly available Coronary Calcium and Chest CT (COCA) dataset (Center, 2022), which contains 790 ECG-gated cardiac CT scans paired with coronary artery calcium (CAC) segmentations in `xml` format. The dataset is fully anonymised; detailed patient demographics are not provided. For each patient, a total Agatston score was computed by summing across per-artery calcium masks. Following clinical convention, we binarise labels at a threshold of 100: patients with scores ≤100 are classified as negative (no/mild CAC), and those >100 as positive (moderate/severe CAC). This threshold was not tuned, but chosen because it reflects the established clinical cutoff between non-actionable and clinically significant CAC (Grundy et al., 2019).

Scans were acquired predominantly at $120\,\mathrm{kV}$ (99.5% of cases) with a slice thickness of $3.0\,\mathrm{mm}$ (98.9%) or $2.5\,\mathrm{mm}$ (1.1%), and in-plane pixel spacing ranging from 0.25 to $0.72\,\mathrm{mm}$ (mean $0.37\,\mathrm{mm}$). CT volumes were resampled to isotropic $\delta x$ mm spacing using trilinear interpolation, where $\delta x$ was matched to the in-plane pixel spacing of the source DICOM. Patients with insufficient slice coverage ($s < 30$) were excluded for quality control, yielding 667 usable scans with a median of 47 slices per volume (range 27–156). Although gated CTs do not capture full thoracic coverage, they represent the cardiac field of view most relevant for coronary artery calcium assessment (Agatston et al., 1990). The cardiac silhouette is

readily identifiable in CXRs, suggesting that analogous regions of interest could be isolated with standard localisation techniques, and that this dataset therefore remains a reasonable surrogate for methodological feasibility testing.

After binarisation at Agatston >100, the dataset comprised 490 negative ($\leq$100) and 177 positive (>100) patients, a ratio of approximately 2.8:1. The full score distribution was: 0–10 ($n$=348, 52.2%), 11–100 ($n$=142, 21.3%), 101–400 ($n$=109, 16.3%), and >400 ($n$=68, 10.2%). Class imbalance was handled via stratified cross-validation.

## 3.2. Digitally Reconstructed Radiographs (DRRs)

Synthetic radiographs were generated directly from CT volumes using the Siddon ray-tracing algorithm, implemented in the open-source `DiffDRR` framework (Gopalakrishnan and Golland, 2022). Siddon projection computes exact line integrals through the CT volume, avoiding interpolation artefacts and preserving small, high-density structures such as calcifications. For each scan, we simulated posterior–anterior (PA) and lateral (LA) projections under a fan-beam geometry with fixed source–detector distance (1085.6 mm) and detector width of 512 pixels at 1 mm spacing; these parameters were selected to ensure full cardiac field-of-view coverage in the resulting projections. Both PA and LA projections used identical source–detector parameters, with the LA view generated by a 90° rotation around the cranio-caudal axis to approximate orthogonal orientation. DRRs were normalised to $[0, 1]$ and resized to $512 \times 512$. This pipeline (Figure 1) yields synthetic radiographs that approximate clinical CXRs while retaining precise CAC labels inherited from the source CTs.

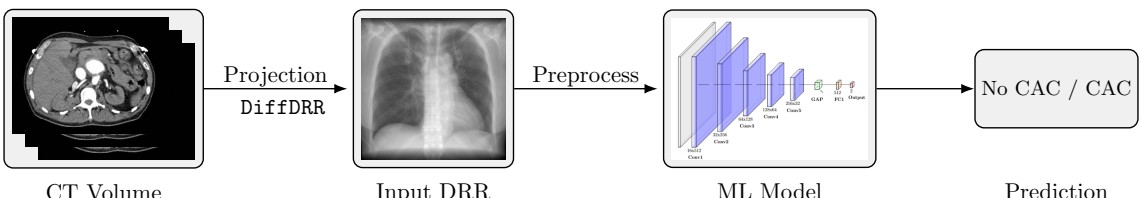

Figure 1: A CT volume (left) is projected into a DRR using Siddon's algorithm (DiffDRR) and fed to a classifier to predict a binary CAC label (Agatston > 100).

## 3.3. Image Enhancement Strategies

To test whether DRRs provide sufficient fidelity for CAC detection—or whether additional enhancements are needed—we evaluate two complementary strategies: **(i)** pre-projection super-resolution and **(ii)** post-projection preprocessing.

### 3.3.1. Pre-projection super-resolution

Native CTs often have anisotropic voxels, where coarse in-plane resolution, from a small number of axial slices, can obscure small calcifications once projected. To test whether resolution recovery enhances CAC depiction, we apply a 4× SRResNet (Ledig et al., 2016)—pretrained on natural images—to sagittal slices prior to projection. The super-resolved slices were then reassembled into a volume, resampled isotropically, and projected identically to

the native-resolution pipeline. We acknowledge that a dedicated CT super-resolution model would be preferable; however, paired low-/high-resolution CT data were unavailable. Medically pretrained super-resolution models remain an avenue for future work. This comparison probes whether fine detail restoration materially improves DRR-based detection.

### 3.3.2. Post-projection image preprocessing

We further test whether DRR adjustments can aid detection by making CAC visually more salient. Three variants were compared: **(1) Original:** unaltered projection. **(2) CLAHE:** contrast-limited adaptive histogram equalisation to locally enhance soft tissue, implemented using an 8×8 tile grid and clip-limit of 2.0, while the unsharp mask used a 5×5 Gaussian kernel ($\sigma$=1.0) with a gain of 1.5. **(3) Calc-focused:** a composite filter designed to highlight calcifications and suppress irrelevant anatomy. Comprised of gamma correction ($\gamma = 1.5$), CLAHE and unsharp masking with a 5×5 Gaussian kernel ($\sigma$=1.0). This axis examines whether heuristic contrast enhancement improves learnability, or whether native DRRs are sufficient (Figure 2).

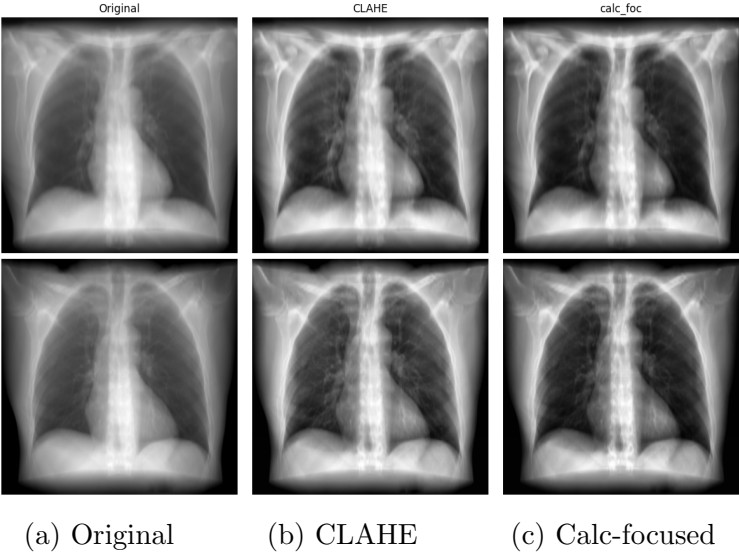

(a) Original      (b) CLAHE      (c) Calc-focused

Figure 2: Preprocessing variants applied to native DRRs (top row) and super-resolved DRRs (bottom row). Columns show (a) Original (unaltered projection), (b) CLAHE (contrast-limited adaptive histogram equalisation), and (c) Calc-focused (gamma correction + CLAHE + unsharp masking).

### 3.4. Experimental Overview

Figure 3 summarises the full experimental design. From each CT volume, DRRs are generated using the native pipeline or after 4× SR upsampling. Each DRR then passes through one of three preprocessing modes. Single-view (PA) models are evaluated across three architectures; multi-view models fuse PA and LA DRRs at three interaction levels. Training strategies (curriculum, SimCLR, or both) are applied to the best-performing fusion configuration.

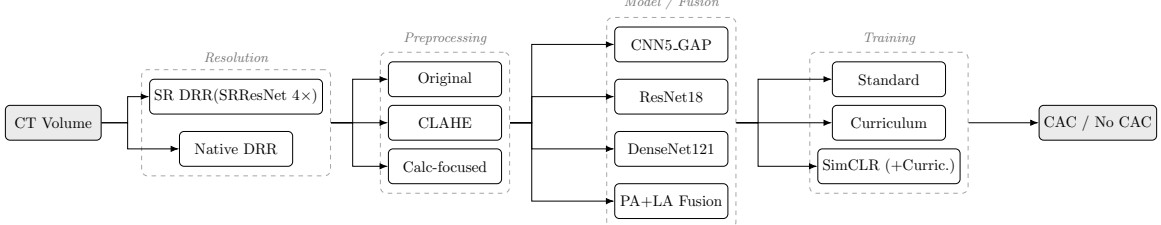

Figure 3: Experimental design overview. Starting from a CT volume, DRRs are generated at native or super-resolved (SR) resolution and subjected to one of three preprocessing modes. Single-view (PA) models and multi-view PA+LA fusion variants are evaluated across training strategies. The best configuration (CNN5_GAP, cross-attention fusion, SimCLR+Curriculum) achieves a mean AUC of 0.754.

## 3.5. Model Architectures

To assess the robustness across model capacity and inductive bias, we evaluate three representative networks: **(i) CNN5_GAP** (Figure 4): a lightweight custom CNN with five convolutional blocks, global average pooling, and a two-layer classifier. Block 1 uses a $5\times5$ kernel; Blocks 2–5 use $3\times3$ kernels; all blocks apply LeakyReLU activations and batch normalisation, followed by dropout ($p$=0.2–0.3) in the classifier head for regularisation. **(ii) ResNet18** (He et al., 2015): a moderate-capacity residual network trained from scratch, serving as a standard baseline. **(iii) DenseNet121** (Huang et al., 2016): pretrained on CheXpert (Irvin et al., 2019) and augmented with a lightweight self-gating spatial attention mechanism derived from CBAM. More specifically, given feature maps $F \in \mathbb{R}^{B \times C \times H \times W}$, we compute a spatial attention mask

$$M = \sigma\left(\frac{1}{C}\sum_{c=1}^{C} F_c\right)$$

where $\sigma(\cdot)$ denotes the sigmoid function. The attended feature map is then $\tilde{F} = F \odot M$, which reweights activations by their spatial salience prior to global pooling.

For multi-view experiments, we use dual-encoder variants incorporating postero-anterior (PA) and lateral (LA) projections. Fusion is tested at three levels: **(i) early fusion** — PA and LA DRRs are channel-concatenated into a single $512\times512\times2$ input before encoding; **(ii) intermediate fusion** — each view passes through a separate encoder and the resulting feature vectors are concatenated before the classifier; **(iii) cross-attention fusion** — a shared encoder processes both views; PA features act as queries attending to LA features as keys and values, allowing the dominant PA signal to selectively incorporate complementary lateral information; attended PA features and original LA features are then combined via a learned sigmoid gate. These designs examine whether DRRs encode complementary view information analogous to real CXR studies.

## 3.6. Training & Evaluation

Models were trained with binary cross-entropy loss and label smoothing ($\varepsilon = 0.1$) using Adam (Kingma and Ba, 2017) (learning rate $10^{-4}$, weight decay $10^{-5}$, batch size 32).

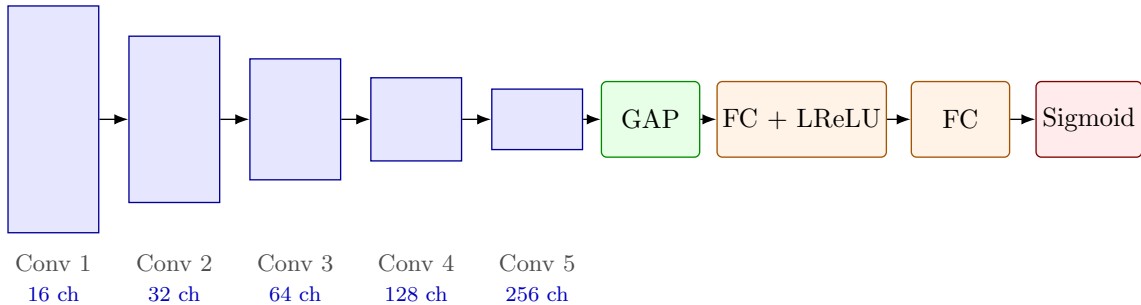

Figure 4: CNN5_GAP architecture. Block height reflects spatial resolution ($512^2 \rightarrow 16^2$, halved per block); channel depth increases $16 \rightarrow 32 \rightarrow 64 \rightarrow 128 \rightarrow 256$. Each block applies Conv (5×5 for Block 1, 3×3 otherwise), batch normalisation, LeakyReLU, and max-pool ↓2×. GAP produces a 256-d vector fed to two FC layers with dropout.

Augmentations consisted of random affine transformations: rotations ($\pm5°$), translations ($\pm5\%$), scaling ($[0.9, 1.1]$) and shearing ($10°$). Horizontal flips were excluded to preserve anatomical context. Early stopping was applied on validation AUC (being the criterion for model selection), with dropout/batch normalisation for regularisation. Multi-view fusion models used a curriculum schedule (Bengio et al., 2009) (extremes first, then borderline), intended as a heuristic to reduce instability from thresholding.

Performance was evaluated with stratified 5-fold cross-validation (CV) across 5 seeds (25 runs) with approximately 530 training and 130 validation DRRs. Metrics included AUC-ROC (primary), with accuracy, precision, recall, and F1 as secondary. All splits were patient-level and stratified by CAC label to avoid data leakage across folds. To reduce optimism bias from volatile early epochs, we discarded the first five epochs and averaged performance over the top five subsequent epochs, selected by highest validation AUC. This conservative reporting may slightly underestimate peak metrics, but ensures robustness. Statistical significance was assessed using paired Wilcoxon signed-rank tests (Hollander et al., 2013) across validation folds. Results were considered significant at $p < 0.05$. Augmentations were applied to 2D DRRs after projection; applying augmentations in 3D before projection would be more physically meaningful and is left for future work. All experiments were conducted on the UNSW Katana high-performance computing cluster, utilising NVIDIA GPU nodes (V100, A100, L40S, H200, and GH200) via a queue-based scheduler.

All preprocessing, training, and evaluation code is implemented in PyTorch and will be made publicly available upon publication for reproducibility.

## 4. Results

We present results in a structured manner, beginning with single-view baselines and ablations on preprocessing and super-resolution, followed by multi-view fusion strategies, and finally training variants such as curriculum learning. Unless otherwise noted, all results are reported as mean validation AUC ($\pm$ standard deviation) across 25 runs, with statistical significance assessed using the Wilcoxon signed-rank test.

### 4.1. Baselines, Preprocessing, and Super-resolution

To assess the impact of preprocessing and super-resolution (SR) on model performance, we evaluated three architectures (CNN5_GAP, DenseNet121, and ResNet18) across three DRR preprocessing modes (Original, CLAHE, Calc-focused) under both native and super-resolved inputs (Table 1). For CNN5_GAP, mean AUCs improved by $+0.015$ ($0.720 \rightarrow 0.735$) from native-original to SR-calc-focused, with SR yielding significant gains under calc-focused and CLAHE preprocessing ($p < 0.05$). DenseNet121 performance degraded ($0.718 \rightarrow 0.688$; $p < 0.05$) under contrast-enhancing preprocessing, with SR offering only minor recovery. ResNet18 remained stable across most modes, with no significant differences observed.

Given its consistent and competitive performance across all conditions, CNN5_GAP was selected as the primary backbone for subsequent experiments.

Table 1: Validation AUC for three architectures (CNN5_GAP; DenseNet121 with CBAM-derived spatial attention; ResNet18) across three preprocessing modes (Original, CLAHE, Calc-focused) and two resolution conditions (Native, SR = $4\times$ super-resolved).[1]

| Preprocessing | CNN5_GAP | | DenseNet121 | | ResNet18 | |
|---|---|---|---|---|---|---|
| | Native | SR | Native | SR | Native | SR |
| Original | $0.720 \pm 0.048$ | $0.727 \pm 0.050$ | $\mathbf{0.718 \pm 0.031^{\dagger}}$ | $0.708 \pm 0.026$ | $0.714 \pm 0.031$ | $0.717 \pm 0.031$ |
| CLAHE | $0.728 \pm 0.049$ | $\mathbf{0.733 \pm 0.043^{*}}$ | $0.688 \pm 0.053$ | $0.695 \pm 0.028$ | $0.716 \pm 0.068$ | $0.702 \pm 0.022$ |
| Calc-foc | $0.729 \pm 0.053$ | $\mathbf{0.735 \pm 0.043^{*}}$ | $0.683 \pm 0.058$ | $0.688 \pm 0.042$ | $0.722 \pm 0.070$ | $0.723 \pm 0.032$ |

### 4.2. Fusion & Training Strategies

We next evaluated whether combining posterior–anterior (PA) and lateral (LA) projections could improve performance over best performing single-view (PA) baseline of $0.735 \pm 0.043$. Three fusion strategies were tested: early fusion (image concatenation), intermediary fusion (latent feature concatenation), and cross-attention fusion. Each was implemented with either shared or unshared encoders and a learnable scalar-gated mechanism to weigh each view's contribution.

Table 2: Validation AUC performance of PA+LA fusion strategies, varying by level of interaction (early, intermediary, or attention) and whether encoders are shared or unshared.

| Fusion Strategy | Shared Encoder | Unshared Encoder |
|---|---|---|
| PA-Only (Baseline) | $0.735 \pm 0.043$ | |
| Early Fusion | $0.702 \pm 0.044$ | – |
| Intermediate Fusion | $0.736 \pm 0.059$ | $0.739 \pm 0.065$ |
| Cross-Attention Fusion | $\mathbf{0.740 \pm 0.043}$ | $0.729 \pm 0.045$ |

As shown in Table 2, early fusion was significantly worse ($p < 0.05$) relative to the PA-only baseline ($0.735 \pm 0.043$). Intermediary fusion achieved mean AUCs of 0.736 (shared

---

1. $\dagger$ $p < 0.05$ vs CLAHE, Calc_foc $*$ $p < 0.05$ vs Original

encoders) and 0.739 (unshared), while cross-attention fusion reached 0.740 (shared) and 0.729 (unshared). Differences among fusion variants were within one standard deviation, and none reached statistical significance *relative* to the PA-only baseline. Notably, LA-only classification (AUC $0.695 \pm 0.043$; see Appendix C) substantially trailed the PA-only baseline.

We also assessed the effect of alternative training strategies on the best-performing fusion model (cross-attention/shared). Specifically, we evaluated curriculum learning, SimCLR-based self-supervised pretraining (Chen et al., 2020), and their combination.

Table 3: Validation AUC across training strategies applied to the best-performing fusion configuration (CNN5_GAP, cross-attention fusion, shared encoders). Results reported as mean $\pm$ std across 25 runs.

| Training Strategy | Mean AUC |
|---|---|
| Fusion (standard) | $0.740 \pm 0.043$ |
| Curriculum Learning | $0.750 \pm 0.046$ |
| SimCLR Pretraining | $0.742 \pm 0.045$ |
| SimCLR + Curriculum | $\mathbf{0.754 \pm 0.055}$ |

Table 3 shows that curriculum learning raised mean AUC from 0.740 to 0.750. SimCLR pretraining yielded 0.742, close to the baseline. SimCLR combined with curriculum achieved the highest observed mean AUC (0.754), though this was comparable within variance. The consistent direction of improvement across folds suggests potential value, but larger datasets are required for confirmation (see Appendix C, Table 8 for pairwise Wilcoxon statistics).

At a clinically relevant operating point of 90% sensitivity, the best-performing model (SimCLR + Curriculum) achieved: precision $0.58 \pm 0.05$, recall $0.91 \pm 0.00$, F1 $0.70 \pm 0.03$, and accuracy $0.63 \pm 0.06$. The low variance in F1 ($\pm 0.03$) demonstrates stable performance across all 25 runs. The modest precision reflects the inherent sensitivity–specificity trade-off and the class imbalance in the dataset.

Performance varied with Agatston score magnitude: accuracy was highest for extreme cases (0–10: $0.74 \pm 0.11$, $n=69$; >400: $0.78 \pm 0.13$, $n=13$) and lowest near the decision boundary (11–100: $0.55 \pm 0.15$, $n=28$; 101–400: $0.72 \pm 0.12$, $n=21$). The reduced accuracy in the 11–100 range is expected, as these patients have subclinical calcification with subtle imaging features, consistent with the weak image-level supervision inherent in binary labels. These results motivate examining how view complementarity and training stability interact under limited data, as discussed next.

## 5. Discussion

Our experiments evaluated the feasibility of using DRRs as a surrogate domain for CAC classification, focusing on: **(i)** fidelity enhancement via super-resolution (SR), **(ii)** signal optimisation through preprocessing, and **(iii)** fusion of complementary projections.

### 5.1. Super-resolution and Preprocessing

Coronary artery calcifications are high-frequency features whose visibility is diminished when CT volumes are reconstructed at low axial resolutions. On its own, SR modestly improved fidelity, yielding consistent but non-significant gains in CNN5 GAP ($+0.007; p = 0.107$). When paired with preprocessing, SR-enabled pipelines achieved improvements over their native baselines ($p < 0.05$), with Calc_foc + SR yielding the strongest single-view performance (0.735). This suggests that SR may play an enabling role: by partially restoring high-frequency structure, it could provide the substrate that preprocessing methods can then amplify into a more discriminative signal.

Preprocessing alone showed model-dependent behaviour. CNN5_GAP adapted well, whereas DenseNet121 degraded significantly ($p < 0.05$). This highlights a key trade-off: pretrained backbones encode strong natural-image priors that are brittle to the contrast shifts introduced by synthetic-domain preprocessing, while lightweight task-specific CNNs trained from scratch are more flexible and can exploit enhanced contrast once fidelity is recovered—consistent with the broader observation that compact models generalise better within surrogate domains where texture and intensity statistics differ from real-world pre-training distributions. However, real CXRs are already acquired at high resolutions; SR's benefit is unique to CT-derived projections. In contrast, preprocessing strategies transfer rather trivially. In clinical CXRs, where acquisition noise and patient variability are greater, local contrast enhancement or normalisation may provide more value by homogenising inputs while highlighting subtle calcifications.

Taken together, these results suggest that fidelity restoration and contrast enhancement jointly determine the effective signal-to-noise ratio for CAC depiction in DRRs.

### 5.2. Fusion & Training Strategies

Given calcifications may be obfuscated in a single projection, we evaluated fusion of PA and LA DRRs. Early image-level fusion performed significantly worse ($p < 0.05$), suggesting destructive interference across modalities. Intermediary feature fusion and cross-attention achieved the best observed gains, though improvements did not reach statistical significance, likely reflecting fold-level variance under limited data. Notably, LA-only classification underperformed (AUC 0.695, see Table 7), indicating that PA projections provide most discriminative information, while the marginal gain from incorporating LA views may be limited under current dataset size and coverage. Curriculum learning stabilised training (+0.010 AUC), SimCLR offered marginal gains, and their combination reached the highest mean AUC (0.754), consistent across folds but not statistically significant. Their complementary effects likely stem from stabilising optimisation (curriculum) and improving representation initialisation (SimCLR). Ultimately, these observations highlight the need for larger, more diverse datasets to fully exploit multi-view complementarity and self-supervised pretraining.

Overall, SR enhances DRR fidelity for lightweight CNNs, preprocessing benefits depend on architecture, and fusion trends are promising but data-limited. DRRs reproduce key behaviours seen in real CXR studies while providing label-rich CT-derived data, substantiating their value as a surrogate domain and motivating future work in domain adaptation.

### 5.3. Failure Mode Analysis

To better understand model behaviour, we performed GradCAM visualisations on representative cases from our best-performing model. Figure 5 shows attention maps for true positives, false positives, and false negatives.

For true positives with high Agatston scores, attention concentrates on the central cardiac silhouette, consistent with the anatomical location of coronary arteries. However, we note this may partially reflect image intensity distribution rather than learned calcification-specific features. False positives show attention shifted toward the spine, suggesting the model may confuse vertebral density with cardiac calcification. False negatives exhibit diffuse, unfocused attention patterns, indicating failure to localise discriminative features in these cases.

These findings suggest that while the model learns anatomically plausible attention for clear positive cases, errors arise from confounding dense structures and insufficient feature localisation for atypical presentations. Future work could incorporate anatomical priors or region-specific losses to improve specificity.

Table 4 further characterises model behaviour by Agatston score range, confirming that classification accuracy is highest for extreme cases and lowest near the decision boundary.

Table 4: Classification accuracy of the best-performing model (SimCLR + Curriculum) stratified by Agatston score range. Results shown as mean $\pm$ std across 25 runs.

| Agatston Range | $n$ | Accuracy |
| --- | --- | --- |
| 0–10 | 69 | $0.74 \pm 0.11$ |
| 11–100 | 28 | $0.55 \pm 0.15$ |
| 101–400 | 21 | $0.72 \pm 0.12$ |
| >400 | 13 | $0.78 \pm 0.13$ |

### 5.4. Comparison to Prior Work

Our best-performing configuration (mean AUC $= 0.754 \pm 0.055$) is numerically comparable to prior CXR-based CAC classification studies. (Kamel et al., 2021) reported an AUC of 0.73 using an attention-augmented VGG16 trained on 1,689 paired CXRs; (D'Ancona et al., 2023) achieved $\approx$0.71 AUC predicting significant coronary artery disease from radiographs; and (Jeong et al., 2024) reported 0.808 AUC using a radiomics approach requiring manual cardiac segmentation. We stress that this comparison is not a direct performance claim: our model is trained on synthetic DRRs with perfect CT-derived labels, whereas prior work contends with real CXR images and their inherent label noise from retrospective CXR–CT pairing. Our result therefore demonstrates *methodological feasibility* rather than superiority.

Our pipeline is **fully automated** (no manual segmentation or handcrafted features) and **intrinsically scalable**: a labelled CT repository yields thousands of precisely aligned DRRs automatically, removing the CXR–CT pairing bottleneck and establishing a reproducible, label-rich platform for future real-CXR transfer.

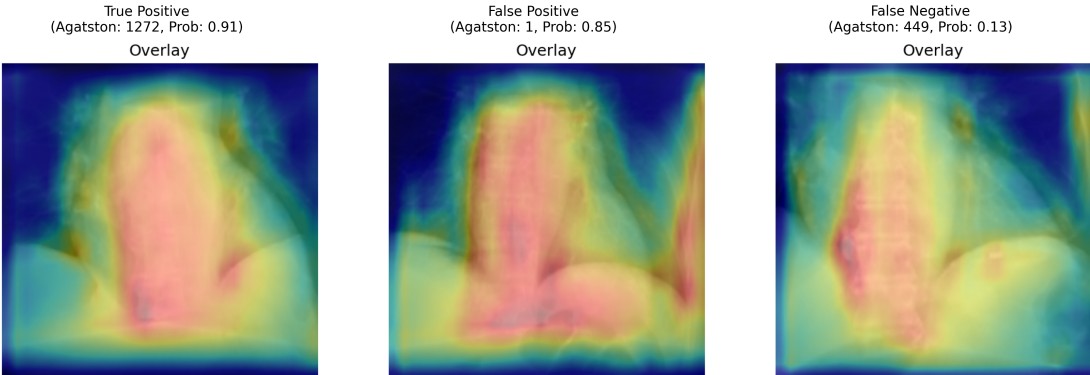

Figure 5: GradCAM visualisations. Left: True positive (Agatston 1272) showing attention on the cardiac silhouette. Centre: False positive (Agatston 1) showing attention shifted toward the spine. Right: False negative (Agatston 449) showing diffuse, unfocused attention.

## 6. Limitations

Several limitations should be noted. Agatston scores provide image-level supervision only; future work should explore per-artery DRR labels to capture spatial heterogeneity. The dataset is small by deep-learning standards ($\sim$130 validation DRRs per fold), contributing to fold-level variance and modest effect sizes. The synthetic domain may not fully replicate clinical CXRs—differences in noise statistics, detector response, and cardiac field-of-view limit direct transfer, so this study demonstrates methodological feasibility rather than clinical readiness; domain adaptation and real-CXR validation remain essential next steps. No systematic hyperparameter search was conducted for CNN5_GAP, and Vision Transformers were excluded due to their higher data requirements. Finally, the absence of a held-out test set means all metrics are cross-validation estimates; a properly held-out test set should be standard practice as datasets scale.

## 7. Conclusion

This study establishes CT-derived DRRs as a viable surrogate training domain for deep learning-based CAC detection. Lightweight CNNs trained from scratch outperform larger pretrained models; pairing super-resolution with contrast enhancement yields statistically significant gains; and curriculum learning stabilises training under label noise, with SimCLR offering only marginal additional benefit. Multi-view fusion produced the best observed performance, though gains did not reach significance under data-limited evaluation. Collectively, these findings demonstrate that DRRs provide a scalable, label-rich foundation for reproducible CAC research prior to real-CXR transfer. With advances in domain adaptation and dataset scale, DRR-pretrained models could underpin population-level cardiovascular risk screening using existing radiography infrastructure.

## Acknowledgments

The authors are grateful for the support by the 2024 (Cardiac, Vascular, and Metabolic Medicine) CVMM THEME COLLABORATIVE GRANT SCHEME.

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

## Appendix A. Statistical Testing Methods

For each paired comparison (same folds/seeds), we apply the Wilcoxon signed-rank test on per-run AUC values. Using 5 randomised seeds across a 5-fold CV, this yields $n = 25$ runs. We report the test statistic and p-value. Significant results are marked as bold at $p < 0.05$ with a $*$.

## Appendix B. Gated Dataset

Table 5: Gated dataset – Wilcoxon signed-rank test within CNN5_GAP across super-resolution and preprocessing modes. Statistical significance at $p < 0.05$ is marked with *.

| Comparison | Stat | $p$-value | Mean AUC A | Mean AUC B |
|---|---|---|---|---|
| Native: orig vs calc_foc | 97.0 | 0.078 | 0.720 | 0.729 |
| Native: orig vs clahe | 104.0 | 0.120 | 0.720 | 0.728 |
| Native: calc_foc vs clahe | 151.0 | 0.771 | 0.729 | 0.728 |
| SR: orig vs calc_foc | 66.0 | **8.07e-03*** | 0.727 | 0.735 |
| SR: orig vs clahe | 89.0 | **0.048*** | 0.727 | 0.733 |
| SR: calc_foc vs clahe | 123.0 | 0.300 | 0.735 | 0.733 |
| Native vs SR (orig) | 129.0 | 0.381 | 0.720 | 0.727 |
| Native vs SR (calc_foc) | 102.0 | 0.107 | 0.729 | 0.735 |
| Native vs SR (clahe) | 91.0 | 0.055 | 0.728 | 0.733 |

Table 6: Gated dataset – Wilcoxon signed-rank test within DenseNet121 across super-resolution and preprocessing modes. Statistical significance at $p < 0.05$ is marked with *.

| Comparison | Stat | $p$-value | Mean AUC A | Mean AUC B |
|---|---|---|---|---|
| Native: orig vs calc_foc | 83.0 | **0.032*** | 0.718 | 0.683 |
| Native: orig vs clahe | 81.0 | **0.028*** | 0.718 | 0.688 |
| Native: calc_foc vs clahe | 122.0 | 0.276 | 0.683 | 0.688 |
| SR: orig vs calc_foc | 66.5 | **0.010*** | 0.708 | 0.688 |
| SR: orig vs clahe | 80.0 | **0.026*** | 0.708 | 0.695 |
| SR: calc_foc vs clahe | 128.0 | 0.367 | 0.688 | 0.695 |
| Native vs SR (orig) | 105.0 | 0.122 | 0.718 | 0.708 |
| Native vs SR (calc_foc) | 150.0 | 0.751 | 0.683 | 0.688 |
| Native vs SR (clahe) | 129.0 | 0.381 | 0.688 | 0.695 |

## Appendix C. Fusion Analysis

### C.1. Lateral-only Performance

Here we assess the strength of the calcium signal in Lateral (LA) projections. Significance was assessed with Wilcoxon signed-rank tests at $p < 0.05$.

Table 7: Wilcoxon signed-rank test results for LA-only AUC across preprocessing modes using CNN5_GAP. Statistical significance at $p < 0.05$ is marked with *.

| Comparison | Stat | $p$-value | Mean AUC A | Mean AUC B |
|---|---|---|---|---|
| LA Native: orig vs calc_foc | 37.0 | **3.29e-04*** | 0.672 | 0.695 |
| LA Native: orig vs clahe | 57.0 | **0.003*** | 0.672 | 0.691 |
| LA Native: calc_foc vs clahe | 120.5 | 0.258 | 0.695 | 0.691 |

### C.2. Fusion and Training Strategy Comparisons

Pairwise Wilcoxon signed-rank test results comparing fusion baselines, curriculum learning, and SimCLR variants. Significant results at $p < 0.05$ are marked with $*$.

Table 8: Pairwise Wilcoxon signed-rank test results comparing fusion baselines, curriculum learning, and SimCLR variants. Statistical significance at $p < 0.05$ is marked with *.

| Comparison | Stat | $p$-value | Mean AUC A | Mean AUC B |
|---|---|---|---|---|
| Early vs Int. Fusion (Shared) | 11.0 | **4.6e-05*** | 0.703 | 0.736 |
| Early vs Int. Fusion (Unshared) | 11.0 | **4.6e-05*** | 0.703 | 0.739 |
| Early vs Cross-Attention (Shared) | 11.0 | **4.6e-05*** | 0.703 | 0.740 |
| Standard vs Curriculum | 132.0 | 0.426 | 0.740 | 0.750 |
| Standard vs SimCLR | 152.0 | 0.791 | 0.740 | 0.742 |
| Standard vs Curriculum + SimCLR | 128.0 | 0.367 | 0.740 | 0.754 |

