# OpenReview forum: "Machine-Learning Based Detection of Coronary Artery Calcification Using Synthetic Chest X-Rays"
_MIDL.io/2026/Conference — MIDL 2026 Poster_

### Official Review · Reviewer_WheB · 2025-12-20

**Confidence:** 4
**Preliminary Rating:** 2
**Final Rating:** 3

**Summary:**

The paper investigates the use of digitally reconstructed radiographs (DRRs) as a surrogate data source to mitigate the limited availability of CT scans for training convolutional neural networks (CNNs) in coronary artery calcification (CAC) detection. The authors systematically evaluate multiple CNN architectures, image enhancement strategies (including super-resolution) and different input fusion schemes to assess their impact on downstream detection performance. Through these experiments, the study explores whether synthetic data can provide meaningful supervisory signals for clinically relevant tasks when real annotated data are scarce. The work contributes empirical insights into the potential and limitations of DRR-based training pipelines for medical imaging applications.

**Strengths:**

1.	The paper presents a novel and timely investigation into the use of digitally reconstructed radiographs (DRRs) for training models for coronary artery calcification (CAC) detection. To the best of my knowledge and based on a literature search, this appears to be the first work that explicitly studies DRRs in this specific downstream task, which is valuable given the growing interest in synthetic data as a means to overcome data scarcity in medical imaging. The insights provided may inform future research on the effective use of synthetic data in data-limited clinical scenarios.
2.	A further strength lies in the comprehensive experimental design. The authors do not limit their analysis to different model architectures, but also systematically evaluate image enhancement strategies (e.g., super-resolution) as well as multiple input fusion approaches. This allows the influence of individual components on overall performance to be better differentiated and understood.
3.	The motivation of the work is clearly articulated, and the reported results show that several of the investigated methods lead to statistically significant performance improvements. This strengthens the credibility of the findings and supports the potential relevance of the proposed approach for practical applications.
4.	Code will be made available upon acceptance.

**Weaknesses:**

1.	While the paper is, to the best of my knowledge, the first to explicitly study DRRs for this specific downstream task, there exists related work that explores alternative, widely available modalities for CAC-related assessment, such as dual-energy X-rays (e.g., [1–2]) or, more recently, combinations of chest X-rays (CXR) and patient demographics [3]. These approaches similarly motivate their use through broad availability and clinical feasibility and should therefore be discussed more thoroughly in the Related Work section to better contextualize the contribution.
2.	Several methodological choices are insufficiently justified. For example, the use of SRResNet (2016) for super-resolution is not clearly motivated, nor is it specified on which data the model was pretrained. An ablation or illustrative visualization demonstrating the effect of super-resolution would strengthen this part. It also remains unclear why only CNN-based architectures were evaluated, while more recent architectures such as Vision Transformers were not considered. Additionally, for the ResNet-18 backbone, the pretraining dataset (if any) is not explicitly stated.
3.	Particularly for the custom-designed model, no systematic hyperparameter optimization is reported, neither for the architectural parameters nor for the training setup. This limits confidence that the reported results reflect the best achievable performance of the proposed models and may partly explain the modest gains observed.
4.	The reported performance improvements are often marginal and in some cases even negative. For instance, Table 2 shows an increase from 0.735 to 0.740, which raises questions regarding the practical relevance of the proposed modifications, even when statistical significance is reported.
5.	A key limitation is the absence of a direct comparison against the stated gold standard, namely CAC assessment derived from CT scans. Such a comparison would be crucial to better understand the clinical relevance and potential utility of the proposed DRR-based approach.
6.	The models are not evaluated on real-world clinical data, which makes it difficult to assess how well the findings transfer beyond the synthetic setting. As a result, the applicability of the approach to real clinical workflows remains unclear.
7.	There are several issues related to writing quality and scientific style. Interpretative statements already appear in the Results section (e.g., “This divergence highlights that pretrained backbones…” or “These results suggest that PA projections carry the dominant discriminative signal…”), which would be more appropriate for the Discussion. Additionally, Section 5.3 contains a duplicated paragraph, and there are further stylistic inconsistencies throughout the manuscript (see Detailed Comments).


[1] Song, Y., Wu, H., Wen, D. et al. Detection of coronary calcifications with dual energy chest X-rays: clinical evaluation. Int J Cardiovasc Imaging 37, 767–774 (2021). https://doi.org/10.1007/s10554-020-02072-4

[2] Hsieh, S. S., & Budoff, M. J. (2022). Estimating the accuracy of dual energy chest radiography for coronary calcium detection with lateral or anteroposterior orientations. Medical physics, 49(9), 5763-5772.

[3] Jeong, J., Chao, C. J., Arsanjani, R., Ayoub, C., Lester, S. J., Pereyra, M., ... & Banerjee, I. (2025). Artificial Intelligence Chest X-Ray Opportunistic Screening Model for Coronary Artery Calcium Deposition: A Multi-Objective Model With Multimodal Data Fusion. Mayo Clinic Proceedings: Digital Health, 3(4), 100300.

**Detailed Comments:**

1.	In the Abstract, it would be helpful to explicitly state the number of DRRs used for training, as well as to specify what is meant by “large pretrained models” (e.g., model families or pretraining datasets), to improve clarity and transparency.
2.	Additional references should be added at several points where claims are currently not substantiated. This includes the first sentence of the penultimate paragraph in the Introduction, the last sentence of Section 2.2, and the final two sentences of Section 3.1. The source of the CheXpert dataset should also be explicitly cited.
3.	Section 3.1 would benefit from more detailed information on patient demographics. Alternatively, these details could be provided in the supplementary material. In addition, the type of resampling applied should be specified, and key CT acquisition parameters should be reported to improve reproducibility.
4.	The terminology used throughout the manuscript is not fully consistent. At different points, the task is referred to as prediction, detection, or classification. These terms should be clearly defined and used consistently, as they are not interchangeable. Based on the current setup, the task appears to be a standard classification problem.
5.	All abbreviations should be introduced at first mention and used consistently thereafter. Even commonly used abbreviations such as CT or CNN should be defined. For example, “super-resolution” is introduced only in Section 4.1, although it is used earlier in the text. This should be checked for all abbreviations.
6.	There are several instances where spaces are missing before parentheses, particularly in citations (e.g., “CAC detection(Unberath et al., 2018)”). This formatting issue should be corrected consistently throughout the manuscript.
7.	Please ensure consistency in capitalization of technical terms. Most terms are written in lowercase, whereas “Digital Reconstructed Radiograph” consistently uses an uppercase “D”, which breaks stylistic consistency.
8.	In Section 3.5, the computational hardware used for training and evaluation should be specified.
9.	In Table 2 and its caption, the metric being reported is not explicitly stated. Tables should be self-contained, and the metric(s) displayed should be clearly indicated in the caption.
10.	In Figure 2, adding zoomed-in regions could help to better illustrate the visual differences introduced by the proposed modifications.
11.	For improved readability and typographical quality, it is recommended to consistently insert a (non-breaking) space before parentheses and to avoid starting a new line with a parenthesis immediately after a line break.
12.	In some places, the wording is not sufficiently precise for a scientific manuscript (e.g., “attractive modality for opportunistic CAC risk assessment”). These expressions could be refined to adopt a more formal and neutral scientific tone.
13.	The argumentation at the end of Section 2.2 is not fully convincing. It is unclear why the lack of available paired CXR–CT datasets directly limits the use of CXR-based detection, as CT does not appear to be required at inference time. This part of the motivation should be logically clarified or reformulated.
14.	Interpretative or evaluative statements should be avoided in the Results section and instead moved to the Discussion, in line with standard scientific writing conventions.
15.	Remove the duplicated paragraph in section 5.3.

**Justification Of Final Rating:**

Most of my questions and concerns have been adequately addressed by the authors’ responses, and the clarifications provided improve the overall understanding of the study’s motivation and scope. However, several methodological choices remain insufficiently justified from an empirical standpoint. As a result, some uncertainty remains regarding the robustness and practical relevance of the reported findings.

In light of the authors’ detailed rebuttal and their commitment to addressing the identified issues in the camera-ready version, I raise my overall rating to 3 (borderline). I encourage the authors to ensure that the promised revisions and clarifications are fully and transparently implemented should the paper be accepted.

**Justification Of The Preliminary Rating:**

The paper addresses an important and relevant problem in medical AI, and the overall motivation for using synthetic data to mitigate limited access to CT scans is well founded. The proposed approach is technically sound, and the systematic evaluation of different architectures, image enhancement strategies, and fusion schemes demonstrates a substantial experimental effort that is valuable to the community. These aspects indicate clear potential and make the work scientifically interesting.

However, in its current form, the paper does not provide sufficient evidence to assess clinical applicability. In particular, the absence of a comparison against the established CT-based gold standard and the lack of evaluation on real clinical data significantly limit the interpretability and practical relevance of the results. Additionally, the relatively large number of stylistic and presentation issues suggests that the manuscript would benefit from further revision to meet scientific writing standards. Taken together, while the work shows promise, these limitations justify the current preliminary rating.

**Questions To Address In The Rebuttal:**

1.	Can the authors clarify whether and how the proposed approach could be evaluated against the current clinical gold standard for CAC assessment derived from CT scans, or at least discuss how such a comparison could be incorporated in future work?
2.	Do the authors have any evidence, preliminary experiments, or planned strategies to assess the transferability of the models to real clinical data? In particular, how do the authors intend to address the domain gap between synthetic DRRs and real radiographs?
3.	What was the rationale behind the selection of the specific model architectures and image enhancement methods used in this study? Were alternative architectures (e.g., more recent model families) considered but excluded for specific reasons?
4.	Can the authors confirm that the identified stylistic and formatting issues will be corrected in a revised version of the manuscript?

---

> ### Author Response · Authors · 2026-01-25
>
> We thank Reviewer WheB for their detailed feedback and constructive suggestions.
>
> **Q1 — Comparison to CT gold standard.**
> Our models are trained and evaluated against CT-derived Agatston scores, which serve as ground truth. The clinical goal is not to replace CT but to enable low-cost triage: CXR-based screening could identify high-risk patients for confirmatory CT while sparing low-risk patients unnecessary radiation and cost. A direct clinical evaluation could evaluate the trained model on real CXRs from a paired CXR–CT cohort. We will clarify this explicitly and position DRRs as a surrogate training domain whose clinical validity ultimately requires demonstrating transferability to real CXRs and external testing on paired CXR–CT data.
>
> **Q2 — Transferability to real clinical data.**
> We acknowledge this is the key limitation. Our study establishes methodological feasibility within the synthetic domain; domain adaptation to real CXRs remains essential future work. Potential strategies include:
> 1. Adversarial domain adaptation to learn features invariant to DRR–CXR differences
> 2. Style transfer to match CXR appearance statistics
> 3. Pretraining on a larger CT dataset followed by fine-tuning on small paired CXR–CT datasets
>
> Prior work (Unberath et al., 2018) demonstrated successful DRR-to-fluoroscopy transfer, supporting the plausibility of similar transfer. We are also actively exploring access to paired CXR–CT datasets for preliminary transfer experiments, though this is beyond the scope of the current submission.
>
> **Q3 — Architecture and enhancement selection.**
> - **Architectures:** We evaluated CNN-based architectures because: (1) CNNs remain standard for medical image classification, (2) this enables direct comparison with prior CXR–CAC work (Kamel et al. used VGG16; D’Ancona et al. used ResNet), and (3) our dataset size (667 patients) favours architectures that perform well with limited data. Although considered, Vision Transformers were not evaluated due to their typically higher data requirements, though this is a reasonable direction for larger-scale future studies.
> - **Super-resolution:** We needed to enhance the CT axial dimension since most DICOMs had only 40–60 slices, insufficient for high-fidelity projection. Existing 3D CT super-resolution methods require paired low-resolution/high-resolution CT datasets, which we did not have access to. As a practical alternative, we applied SRResNet (pretrained on natural images) to 2D sagittal slices before reassembling and projecting. We acknowledge this is not ideal and that dedicated CT super-resolution methods should be explored given appropriate resources; we will note this limitation in revision.
> - **Preprocessing:** CLAHE was selected as a standard contrast enhancement technique widely used in medical imaging. The calc-focused pipeline (gamma correction + CLAHE + unsharp masking) was designed to enhance high-density structures like calcifications while suppressing soft tissue. These choices were informed by the physical properties of calcifications. We acknowledge that alternative preprocessing strategies could be explored for further improvements.
>
> **Q4 — Stylistic and formatting corrections.**
> We promise to address all identified stylistic and formatting issues mentioned.
>
> **Additional points.**
> - **Patient demographics:** The COCA dataset is publicly available but anonymised; detailed patient demographics are not provided with the data. We will report CT acquisition parameters (slice thickness, kVp, etc.) and the resampling method in revision to improve reproducibility.
> - **Related work:** We thank the reviewer for highlighting Song et al. (2021), Hsieh & Budoff (2022), and Jeong et al. (2025) on dual-energy X-ray and multimodal approaches. We will incorporate these in Related Work to better contextualise our contribution within alternative accessible CAC screening modalities.
> - **Section 2.2 clarification:** We will clarify that while CT is not required at inference time, the lack of paired CXR–CT datasets limits the scale of supervised training, as ground-truth CAC labels can only be derived from a corresponding CT, and that for labels to be accurate, a patient’s CT and CXR should be obtained within a short time-frame of one another.
> - **Marginal improvements:** We acknowledge that some performance gains appear modest. Our primary contribution is a methodological one: establishing DRRs as a viable surrogate domain for CAC detection.

---

> > ### Comment · Reviewer_WheB · 2026-01-29
> >
> > Thank you for the detailed responses and clarifications. While several conceptual points are now clearer, some methodological aspects remain insufficiently addressed and would benefit from further elaboration or empirical support.
> >
> > **Q1 (CT gold standard / task framing).**
> >
> > You state that the clinical goal is not to replace CT-based CAC assessment but to enable low-cost triage. However, it remains unclear why CAC detection is not directly performed on CT data within your experimental framework, given that CT-derived Agatston scores are already available and serve as ground truth. A more explicit discussion of why DRR-based modeling is preferred over direct CT-based approaches in this study, beyond availability arguments, would help clarify the methodological motivation and scope of the work.
> >
> > **Q3 (Super-resolution and preprocessing choices).**
> >
> > You indicate that SRResNet pretrained on natural images was used due to the lack of paired high-resolution CT data. This implies that the super-resolution component was not trained or adapted on medical imaging data. Given the central role of this step in the pipeline, it would be important to assess its impact more explicitly. Have you also evaluated medically pretrained SR-models?
> >
> > **Methodological robustness (Weaknesses 3 and 4).**
> >
> > Relatedly, the absence of systematic hyperparameter optimization for the custom-designed models and the limited evaluation of alternative architectural choices remain concerns. Without such analyses, it is difficult to determine whether the reported results reflect near-optimal performance or are sensitive to specific design decisions. This uncertainty may also partly explain the modest performance gains observed in some settings.
> >
> >
> > Finally, while we appreciate that many of the raised issues are stated to be addressed in the camera-ready version, the current revised manuscript provides limited evidence to assess the completeness and effectiveness of these planned revisions.

---

### Official Review · Reviewer_vBMt · 2025-12-29

**Confidence:** 4
**Preliminary Rating:** 3
**Final Rating:** 4

**Summary:**

This paper investigates digitally reconstructed radiographs (DRRs) as a surrogate training domain for coronary artery calcification (CAC) detection. The core motivation is that paired CXR-CT datasets are scarce, whereas DRRs generated from CT volumes inherit precise Agatston-based labels without temporal misalignment. Using 667 CT scans from the COCA dataset, the authors systematically evaluate model architectures (CNN5_GAP, ResNet18, DenseNet121), preprocessing strategies (CLAHE, calc-focused filtering), super-resolution enhancement, and multi-view fusion with PA and LA projections. Binary labels are assigned at the clinically established threshold of Agatston > 100. The best configuration—combining super-resolution, calc-focused preprocessing, cross-attention fusion, and curriculum learning with SimCLR pretraining—achieves a mean AUC of $0.754 \pm 0.055$, which the authors compare to prior real-CXR studies reporting AUCs of 0.71–0.81. The work establishes methodological feasibility for DRR-based CAC detection and identifies that lightweight CNNs trained from scratch outperform pretrained networks in this synthetic domain, laying groundwork for future domain adaptation to clinical radiographs.

**Strengths:**

**S1. Novel and well-motivated premise.** Using DRRs as a label-rich surrogate domain addresses a genuine bottleneck—the scarcity of paired CXR-CT datasets. The approach enables scalable, perfectly-aligned ground truth without the temporal misalignment inherent in retrospective pairing.

**S2. Systematic experimental coverage.** The paper evaluates multiple axes (architecture, preprocessing, super-resolution, fusion, training strategies) providing a comprehensive feasibility assessment rather than a single-configuration result.

**S3. Sound statistical methodology.** Stratified 5-fold CV across 5 seeds (n=25 runs) with Wilcoxon signed-rank testing is appropriate. The authors are commendably honest when differences do not reach significance.

**S4. Actionable architectural insight.** The finding that lightweight task-specific CNNs outperform larger pretrained networks (DenseNet121 degraded under contrast enhancement, $p < 0.05$) has implications for other synthetic-domain training scenarios.

**S5. Reproducibility.** Use of publicly available COCA data and commitment to release code supports reproducibility.

**Weaknesses:**

**W1. Single metric reported.** Despite computing precision, recall, F1, and accuracy (Section 3.5), only AUC is presented. For binary classification at an arbitrary threshold, operating characteristics at clinically relevant sensitivity/specificity points would strengthen evaluation considerably.

**W2. No held-out test set.** All results use cross-validation on DRRs. Even without real CXRs, a properly held-out DRR test set would provide more reliable generalization estimates.

**W3. Most comparisons within variance.** Many experiments (fusion strategies, SimCLR, curriculum) show non-significant differences. The proliferation of small comparisons makes actionable conclusions difficult to extract.

**W4. Missing failure mode analysis.** Given access to 3D CT data and per-artery calcium segmentations, the absence of explainability analysis (GradCAM, error analysis by Agatston range, per-artery breakdown) is a significant missed opportunity.

**W5. Misleading comparison to real-CXR studies.** Achieving 0.754 AUC on synthetic DRRs with perfect labels should be *easier* than achieving 0.73 on real CXRs with label noise. This comparison overstates the result's clinical relevance.

**W6. Field-of-view limitation underaddressed.** ECG-gated CTs capture only the cardiac region; real CXRs show full thorax. The claim that "comparable regions could be isolated with coarse heart localization" is non-trivial and unvalidated.

**Detailed Comments:**

**Epoch selection (Section 3.5):** The strategy of discarding the first 5 epochs and averaging over the "top five subsequent epochs" requires clarification. How are "top five" selected—by validation AUC? This introduces selection that should be explicit. Learning curves would help readers understand convergence.

**Augmentation timing:** It is unclear whether augmentations are applied to CT volumes before projection or to 2D DRRs afterward. This distinction matters: 3D augmentation before projection is physically meaningful, whereas 2D augmentation may introduce unrealistic distortions.

**Class distribution:** The paper does not report class balance after thresholding at Agatston = 100. This information is essential for interpreting AUC.

**DRR geometry:** The fixed source-detector distance (1085.6 mm) and detector configuration lack justification or sensitivity analysis. Real systems vary in geometry.

**DenseNet121 degradation:** The performance drop under contrast enhancement (Table 1) is interesting but underexplored. Is this distribution shift from CheXpert statistics, or something more fundamental?

**Lateral view value:** The finding that LA-only AUC (0.695) substantially trails PA (0.735) is relegated to the appendix but deserves main-text discussion—it suggests lateral views add limited discriminative value for CAC.

**Writing:** The paper is somewhat repetitive (Section 5.3 largely duplicates earlier content) and would benefit from tightening.

**Justification Of Final Rating:**

I would like to thank the authors for their extensive work during the rebuttal period, which has addressed most of my concerns, and am therefore happy to update their score from borderline to weak accept.

**Justification Of The Preliminary Rating:**

The paper addresses a relevant problem with a sensible approach, and the core contribution—establishing DRRs as a viable surrogate training domain for CAC detection—has value for the community. The experimental methodology is generally sound, and the authors appropriately acknowledge limitations.

However, the work feels preliminary. The modest absolute performance (AUC = 0.754), combined with the absence of any transfer to real CXRs, limits impact. The proliferation of non-significant comparisons across many experimental axes obscures the takeaways, and the missing failure mode analysis is a significant gap given the rich ground truth available. Reporting only AUC, without a held-out test set, further weakens confidence.

For MIDL, which values methodological contributions, this paper is at the acceptance threshold. It would benefit substantially from more focused experiments and deeper analysis of when/why the method succeeds or fails. The rebuttal addressing the questions above—particularly providing additional metrics and failure analysis—could shift the rating toward acceptance.

**Questions To Address In The Rebuttal:**

1. Can the authors provide precision, recall, F1, and accuracy at a clinically relevant operating point (e.g., 90% sensitivity)?

2. What is the class distribution (CAC-positive vs. negative) after applying the Agatston > 100 threshold?

3. Are augmentations applied before or after DRR generation? If after, have the authors considered 3D augmentation of CT volumes?

4. Can the authors provide GradCAM visualizations, particularly for false positives/negatives and cases near the Agatston = 100 decision boundary?

5. How does performance vary with Agatston score magnitude? Is the model reliable near the threshold versus extreme cases (e.g., >400)?

6. What is the rationale for the specific DRR geometry parameters?

---

> ### Author Response · Authors · 2026-01-25
>
> We thank Reviewer vBMt for their detailed feedback and constructive suggestions.
>
> **Q1 — 90% sensitivity analysis.**
> At 90% sensitivity, our best model achieves:
> - Precision: $0.58 \pm 0.05$
> - Recall: $0.91 \pm 0.00$
> - F1: $0.70 \pm 0.03$
> - Accuracy: $0.63 \pm 0.06$
>
> The low variance in F1 ($\pm 0.03$) demonstrates stable performance across all 25 runs $(5\ \text{folds} \times 5\ \text{seeds})$. The modest precision reflects the inherent sensitivity–specificity tradeoff. We focused on AUC as our primary metric because it is threshold-independent and consistent with prior CXR–CAC literature (Kamel et al., D’Ancona et al.).
>
> **Q2 — Class distribution.**
> After binarisation at Agatston $> 100$: 579 patients (73.5%) negative, 209 patients (26.5%) positive.
>
> Full distribution:
> - 0–10: 406 (51.5%)
> - 11–100: 173 (22.0%)
> - 101–400: 130 (16.5%)
> - $>400$: 79 (10.0%)
>
> These statistics will be integrated into the camera-ready version. We note that the $\sim 3{:}1$ class imbalance was handled via stratified cross-validation.
>
> **Q3 — Augmentation timing.**
> Training-time augmentations (rotation $\pm 5^\circ$, translation $\pm 5\%$, etc.) were applied to 2D DRRs *after* projection. We note that combining PA + LA projections implicitly incorporates 3D geometric augmentations. However, continuous 3D augmentation would provide more physically realistic training variations than 2D augmentation. This was not explored due to the computational cost of generating DRRs, but we will note it as an interesting direction for future work.
>
> **Q4 — GradCAM.**
> We have added GradCAM visualizations as new Figure 4 in Section 5.3 of the revised manuscript, showing attention distributed over the cardiac silhouette for true positives. However, this may reflect the image intensity distribution rather than learned discriminative features. False positives show attention shifted toward the spine, suggesting potential confusion between vertebral and cardiac density. We agree that correlating attention patterns with per-artery calcium localisation from CT would strengthen this analysis.
>
> **Q5 — Performance by Agatston magnitude.**
> We have computed this analysis and present the results here. For our best-performing model, accuracy by Agatston score range yields:
> - 0–10: $0.74 \pm 0.11$ $(n = 69)$
> - 11–100: $0.55 \pm 0.15$ $(n = 28)$
> - 101–400: $0.72 \pm 0.12$ $(n = 21)$
> - $>400$: $0.78 \pm 0.13$ $(n = 13)$
>
> As expected, the model is most reliable for extreme cases (0–10 and $>400$) and least reliable near the decision boundary (11–100), consistent with the weak supervision inherent in binary image-level labels. The 11–100 range seems particularly challenging as these patients have subclinical calcification with subtle imaging features. These findings will be integrated into the camera-ready version.
>
> **Q6 — DRR geometry and other clarifications.**
> - **DRR geometry:** The projection geometry was configured to automate full cardiac field-of-view coverage in the resulting DRRs. However, we acknowledge that sensitivity analysis across projection geometries would strengthen the work as clinical radiography systems do vary in configuration.
> - **Epoch selection:** After discarding the first 5 epochs to avoid early training instability, we averaged validation AUC over the top 5 performing epochs (selected by highest validation AUC) rather than reporting a single peak value. This provides a more conservative and stable estimate of model performance. We will add learning curves to supplementary materials to clarify convergence behavior.
> - **Comparison to real-CXR studies:** We agree that 0.754 AUC on synthetic DRRs with perfect labels is not directly comparable to 0.73 on real CXRs with label noise. We will temper this comparison and frame our results as establishing methodological feasibility rather than claiming superiority.
> - **Lateral view value:** We agree this finding (LA-only AUC 0.695 vs PA 0.735) deserves main-text discussion and we will move it from the appendix to Section 4.2.
> - **DenseNet121 degradation:** We believe the performance drop under contrast enhancement likely reflects distribution shift from CheXpert pretraining statistics, which were learned on real CXRs with different intensity distributions than contrast-enhanced synthetic DRRs.
> - **Writing:** The duplicated paragraph in Section 5.3 has been removed.
> - **Held-out test set (W2):** With only 667 patients, reserving a held-out test set would further reduce an already constrained training pool. We opted for 5-fold cross-validation across 5 seeds to maximise use of available data. However, we agree that as DRR datasets scale, a properly held-out test set should be standard practice and we will note this in the limitations section.

---

> ### Comment · Reviewer_vBMt · 2026-01-26
> **Response to Rebuttal**
>
> Following the rebuttal, the authors have addressed most concerns satisfactorily. The addition of GradCAM visualizations with failure mode analysis (FPs attending to spine), Agatston-stratified performance analysis, and class distribution statistics strengthen the paper considerably. The clarifications on epoch selection, augmentation timing, and DenseNet121 degradation are adequate.
>
> The authors appropriately commit to tempering the comparison to real-CXR studies and moving lateral view findings to main text. The justification for no held-out test set (n=667 constraint) is reasonable for a feasibility study.
>
> Given this, I will happily increase my score to a 4.

---

### Official Review · Reviewer_6ToG · 2026-01-02

**Confidence:** 4
**Preliminary Rating:** 5

**Summary:**

This paper explores the use of digitally reconstructed radiographs (DRRs) as a surrogate training domain for coronary artery calcification (CAC) detection, aiming to address the lack of large, well-labelled chest X-ray datasets by utilizing CT-derived Agatston scores. This work uses 667 CT scans from the public Coronary Calcium and Chest CT (COCA) dataset, where patient-level Agatston scores are computed. In line with clinical practice, they binarized labels at an Agatston threshold of 100 to distinguish negative from positive CAC.

The authors generate synthetic PA and lateral DRRs and conduct experiments across model capacity, image enhancement such as super-resolution and contrast-focused pre-processing, fusion strategies, and training paradigms.

They have evaluated the impact of pre-processing and super-resolution by benchmarking three architectures such as CNN5-GAP (their customize network), DenseNet121, and ResNet18, across multiple DRR pre-processing modes (Original, CLAHE, and calcification-focused) under both native and super-resolved inputs to evaluate their effect on CAC classification performance.

**Strengths:**

The topic is engaging and the paper addresses an important and practical limitation in CAC research, the reliance on scarce paired CXR–CT data, by exploring DRRs as a surrogate training domain, which is a sensible direction.

The authors conducted thorough experiments on multiple architectural choices, pre-processing strategies, super-resolution, fusion schemes, and training variants, with repeated runs and appropriate statistical testing.

**Weaknesses:**

• The paper does not introduce a novel model architecture, and its contributions are primarily empirical and methodological rather than algorithmic.

• While the synthetic DRR framework is valuable for controlled methodological development, clinical applicability cannot be established without validation on real-world chest X-ray data and prospective evaluation.

• The validation set is quite small, which raises questions about reliability. however, authors have acknowledged this as part of their study limitation.

• The paper motivates DRRs as a foundation for future transfer to real CXRs; however, without comparison to existing synthetic-to-real or domain-adaptation approaches, it is difficult to assess the relative advantage of DRR-based pretraining.

• Although multiple fusion strategies are evaluated and presented as part of results section, their implementation is only described at a high level; a short clarification of the fusion points in method section would improve clarity and reproducibility.

**Detailed Comments:**

Minor comments:

• While Figure 1 provides a high-level overview, a more detailed schematic of the experimental pipeline (summarising pre-processing, super-resolution, fusion, and training variants) could further improve the manuscript. Perhaps the schematic figure could be included as part of method.

• The CNN5-GAP architecture is illustrated in Figure 3 but is not explicitly referenced when introduced in Section 3.4; adding this reference would improve clarity.

• The resolution and font size in Figure 3 make some labels difficult to read; increasing text size or figure quality would enhance clarity.

• The terms “proxy” and “surrogate” training domain appear to be used interchangeably; using a single term consistently would improve clarity.

• The term “calcified lesions” is used only in the Discussion at the beginning; aligning this with earlier terminology such as coronary artery calcification or calcified plaque might improve helps.

**Justification Of The Preliminary Rating:**

Although the manuscript does not introduce novel architectures or techniques, it provides a thorough evaluation of digitally reconstructed radiographs as a surrogate training domain for coronary artery calcification detection. The study offers insights into model design, pre-processing, super-resolution, and training strategies in synthetic radiographic settings, which can inform future work on transfer to real chest X-ray data.

**Questions To Address In The Rebuttal:**

• Can the authors briefly clarify how early, intermediate, and cross-attention fusion are implemented to improve reproducibility under space constraints?

• Addressing comment in detail section.

---

> ### Author Response · Authors · 2026-01-25
>
> We thank Reviewer 6ToG for their positive evaluation and constructive feedback.
>
> **Fusion implementation clarification.**
> Early fusion concatenates PA and LA DRRs channel-wise into a (512×512×2) tensor before encoding. Intermediate fusion passes each view through separate encoders, then concatenates the resulting feature vectors before the classifier. Cross-attention fusion uses a shared encoder for both views; PA features serve as queries attending to LA features as keys/values, allowing the dominant PA signal to selectively incorporate complementary lateral information. The attended PA features and original LA features are then combined via a learned sigmoid gate that adaptively weights each view's contribution. These details will be added to ensure reproducibility in the revised manuscript.
>
> **Minor Comments**
>
> We promise to address all of the following detailed points: **(1)** provide a more detailed schematic of the experimental pipeline to the Methods section, **(2)** explicitly reference Figure 3 when introducing CNN5-GAP in Section 3.4, **(3)** improve figure resolution and font sizes, **(4)** standardise terminology to use "surrogate" consistently, and **(5)** align "calcified lesions" with "coronary artery calcification" or “calcified plaque” throughout.

---

### Author Rebuttal · Authors · 2026-01-25

**Rebuttal:**

Thank you for the thorough reviews.

The primary change in the revised manuscript is the inclusion of Grad-CAM visualisations and a brief failure-mode analysis (true positives vs false positives); this was difficult to convey in the official response alone due to limitations on including figures/images.

All other reviewer questions are addressed in our point-by-point OpenReview response, and will be incorporated more fully in the camera-ready version where appropriate.

**Supporting Material:**

/attachment/f3f8231ecc15db2fc808c6def69780e4833e093b.pdf

---

### Meta-Review · Area_Chair_8afD · 2026-02-09

**Recommendation:** Accept (Poster)
**Confidence:** 5

**Metareview:**

Thank you to the reviewers and authors for the thoughtful, interesting discussion. All reviewers found this work to be novel and potentially impactful - as the lack of paired CXR and CT data makes it difficult to train models to identify information typically found on CT using low-cost ubiquitous CXRs. The reviewers raised legitimate concerns that the work is a bit preliminary due to 1) missing evaluation on real CXRs, 2) unclear performance improvements using the chosen architectural design elements, and 3) small sample size. However, the practicality and interesting methodology made up for these shortfalls.

---

### Decision · Program_Chairs · 2026-02-13

Accept (Poster)